# Telenomus nizwaensis (Hymenoptera: Scelionidae), an important egg parasitoid of the pomegranate butterfly Deudorix livia Klug (Lepidoptera: Lycaenidae) in Oman

A. Polaszek[1]*, A. Al-Riyami[2,3], Z. Lahey[4], S. A. Al-Khatri[3], R. H. Al-Shidi[5], I. C. W. Hardy[2¤]

1 Department of Life Sciences, Natural History Museum, London, United Kingdom, 2 School of Biosciences, University of Nottingham, Nottingham, United Kingdom, 3 Directorate General of Agricultural Development, Ministry of Agriculture, Fisheries and Water Resources, Muscat, Sultanate of Oman, 4 Department of Evolution, Ecology and Organismal Biology, The Ohio State University, Columbus, Ohio, United States of America, 5 Plant Protection Research Centre, Directorate General of Agricultural and Livestock Research, Ministry of Agriculture, Fisheries and Water Resources, Muscat, Sultanate of Oman

¤ Current address: Department of Agricultural Sciences, University of Helsinki, Helsinki, Finland
* a.polaszek@nhm.ac.uk

## Abstract

The pomegranate butterfly Deudorix (= Virachola) livia is the major pest of pomegranate, a crop of economic importance, in Oman. A species of parasitoid wasp in the hymenopteran family Scelionidae is responsible for high levels of mortality of its eggs. This wasp is described herein as Telenomus nizwaensis Polaszek **sp. n.**, based on morphology and DNA sequence data. T. nizwaensis is currently known only from D. livia, which is also a pest of economic importance on other crops in North Africa, the Arabian Peninsula, and the Mediterranean. We summarise current knowledge of T. nizwaensis life-history and its potential to provide biological pest control.

## Introduction

Pomegranate Punica granatum L. (Lythraceae, Punicoideae) is a fruit-bearing deciduous shrub or small tree that originates from the Near-East, where it has been cultivated for centuries [1]. It is increasingly important in other parts of the world, such as China, SE Asia, and the Americas [2–4]. In Oman, pomegranate is grown mainly in the mountainous region of Al Jabal Al Akhdar (near the city of Nizwa, Ad Dakhiliyah Governorate), with a total of more than 22,000 productive trees [5]. They are grown at ca. 3000m, where the temperature averages 24˚C during the summer fruiting season [6]. Mature fruits are both sweet and large, and in Oman can sell for the equivalent of US$5 per fruit in some seasons. Hence, pomegranate cultivation is a major aspect of the agricultural economy [7]. Unfortunately, during each growing season Omani pomegranates are infested by a lepidopteran pest called the pomegranate butterfly, Deudorix (= Virachola) livia Klug (Lepidoptera: Lycaenidae) [8], and this causes damage of economic importance. The larval stage of D. livia burrows within the fruit, causing spoilage of

**Data Availability Statement:** The data underlying this study are available on Dryad (https://doi.org/10.5061/dryad.sxksn0329).

**Funding:** A. A-R: PGE055580 D.P.S/321/2017 Sultanate of Oman, Ministry of Higher Education, Research and Innovation https://www.mohe.gov.om. We thank the Anglo-Omani Society, London UK, for assistance with publication charges. The funders had no role in study design, data collection and analysis, decision to publish, or preparation of the manuscript.

**Competing interests:** The authors have declared that no competing interests exist.

seeds, subsequent infection by saprophytic bacteria and growth of moulds, rotting of part or all of the fruit, and, in some cases, the fall of the fruit.

Attempts to control *D. livia* in Oman include an annual IPM program using physical (fruit bagging), chemical control (1–2 applications of Deltamethrin during high infestation periods) and mainly biological control using parasitoid wasps. *Trichogramma* species (Chalcidoidea: Trichogrammatidae) have been imported for release, and a species of *Telenomus* (Platygastroidea: Scelionidae) is naturally present in the Al Jabal Al Akhdar mountains. Specimens of these *Telenomus* were sent to the first author (AP) in 2008, and examined morphologically, including the preparation of male genitalia, critical for recognising *Telenomus* at species-level. The conclusion based on morphology alone was that these specimens belong to an undescribed species. Because of the difficulty in unequivocally establishing them as a species new to science on the basis of morphology alone, they were not formally described at that time. With the advent of relatively rapid and affordable techniques of DNA extraction and sequencing, it recently became feasible to present a formal description, using an integrated taxonomic approach. Specimens from the same host and locality as the initial sample were sent to the first author in 2018. These underwent a DNA extraction protocol that leaves the sclerotized structures intact, and the resulting DNA sequence data supported the initial conclusion that the specimens belonged to an undescribed species. Here we describe this new species as *Telenomus nizwaensis* Polaszek, both to facilitate future identification, and to provide the formal nomenclature essential to support further work using this parasitoid as a biological control agent. The name of this new species is derived from Nizwa, the ancient capital of Northern Oman, about 60 km from Al Jabal Al Akhdar. Further, we provide a summary of its currently known biology and areas for further research.

## Materials and methods

### Specimen repositories: Abbreviations

Individual specimen numbers are given below under "Material examined". Specimens are permanently publicly deposited in the following repositories:

NHMUK: Natural History Museum, London UK.

ONHM: Oman Natural History Museum, Muscat, Oman.

USNM: National Museum of Natural History, Smithsonian Institution, Washington D.C., USA.

### Morphological study

Specimens were obtained from the field by rearing from egg masses of *D. livia* (R.A. and A.A., unpublished data). Because the specimens were collected in Oman by Omani researchers from the Ministry of Agriculture, Fisheries and Water Resources, no collecting permits were required for the described study. During the pomegranate fruiting season (April-September), eggs were collected from fruits, leaves and occasionally from stems. They were placed in Eppendorf tubes and transferred to the laboratory in a cool box (to avoid desiccation). In the laboratory, eggs were kept at a maximum of 5 eggs per tube to limit crowding. When adult parasitoids emerged, around three weeks later, they were stored in a new tube in 100% ethanol. Card-mounted specimens were observed with a Leitz binocular microscope at magnifications ranging from 10× to 40×. Slide-mounted structures were observed with a Leitz Dialux 20 EB compound microscope at magnifications ranging between 40× and 400×. Several specimens were gold-palladium coated and photographed with a Zeiss Ultra Plus field emission Scanning Electron Microscope at magnifications between 300× and 900×. Images were generated as follows: Light microscope images: Canon DSLR with 10× Mitutoyo objective, processed with

HeliconFocus stacking software; Compound microscope images (slide-mounted structures): Leitz Dialux 20EB compound microscope using Nomarski Differential Interference Contrast (DIC) illumination, photographed with MicroPublisher 5.0 RTV camera; scanned sections stacked and combined using Synoptics AutoMontage® software; Scanning electron micrographs: Zeiss Ultra Plus field emission Scanning Electron Microscope. All final image editing was with Adobe Photoshop CC®. Morphological terminology largely follows Johnson [9] and Talamas et al. [10]; genitalia terminology follows Johnson [9] and Polaszek and Kimani [11]. The holotype of *T. nizwaensis* is deposited in NHMUK, paratypes in NHMUK, ONHM and USNM.

### DNA sequencing

Eight *Telenomus nizwaensis* individuals from Oman (4 females, 4 males) were subjected to "non-destructive" DNA extraction. Genomic DNA was extracted using established protocols [12, 13], which leave the sclerotized parts of the specimen intact. Specimens were then critical point dried and card-mounted, with selected individuals then dissected and mounted in Canada balsam on microscope slides, and others gold-palladium coated for SEM examination.

To generate CO1 sequences, the standard "barcode" forward primer LCO1490 [14] was paired with the reverse primer C1-N-2329 [15], resulting in an amplicon that is longer than the standard "barcode" region by about 160 bp. The PCR cycle for the 5′ end of the CO1 consisted of an initial denaturation step of 94˚C for 2 min, followed by 40 cycles of 94˚C for 30 s, 40˚C for 60 s and 72˚C for 30 s, and a final extension step of 10 min at 72˚C. For 28S the conditions where similar except for annealing at 55˚C for 30 s.

The 28S D2 fragment was amplified with the primers D23F (5′–GAG AGT TCA AGA GTA CGT G–3′) [16] and D2R (aka 28S-Rev) (5′–TTG GTC CGT GTT TCA AGA CGG–3′) [17]. All reactions were carried out in a 25μl reaction volume containing 5μl of template DNA, 2.5μl of 10× PCR buffer, 0.75μl of 50 mM MgCl2, 0.2μl dNTPs solution (25 mM each), 1.25μl of each primer (10μM), 0.3μl Taq polymerase (5u/μl Biotaq, Bioline) and PCR grade water to final volume.

Both DNA strands were sequenced at the Natural History Museum Life Sciences DNA Sequencing Facility (London) using the same primers used for the PCR. Forward and reverse sequences were assembled and edited in Sequencher version 4.8. Contigs of the 28S rDNA and COI genes were then assembled using GeneStudio Professional Edition (v. 2.2.0.0).

### Phylogenetic analyses

Phylogenetic analyses were conducted using the CO1 and 28S sequences of three *T. nizwaensis* specimens (specimens subsequently designated male holotype and 2 male paratypes) and additional platygastroid sequences retrieved from GenBank. Sequence alignments of both genes were generated with MAFFT (v. 7.394) [18] using the L-INS-i algorithm.

Phylogenies were estimated for 28S, COI, and 28S+COI in IQ-TREE (v. 1.6.12) [19]. The best nucleotide substitution model for each gene was selected with ModelFinder [20]. The combined analysis (28S+CO1) was run with two partitions, one for each gene, using the best nucleotide substitution model identified in the previous step [21]. Branch support was estimated with 1000 ultrafast bootstrap replicates [22]. The trees presented are those with the best log-likelihood score from 100 independent runs. *Trissolcus thyantae* Ashmead (Scelionidae: Telenominae) was selected as the outgroup in each analysis based on the results of Taekul et al. [23].

## Nomenclatural acts

The electronic edition of this article conforms to the requirements of the amended International Code of Zoological Nomenclature, and hence the new names contained herein are available under that Code from the electronic edition of this article. This published work and the nomenclatural acts it contains have been registered in ZooBank, the online registration system for the ICZN. The ZooBank LSIDs (Life Science Identifiers) can be resolved and the associated information viewed through any standard web browser by appending the LSID to the prefix "http://zoobank.org/". The LSID for this publication is: urn:lsid:zoobank.org:pub:CE45C3B6-038B-4-9EA5BA8254BCORRECT. The electronic edition of this work was published in a journal with an ISSN and has been archived and is available from the following digital repositories: PubMed Central, LOCKSS.

## Description

***Telenomus nizwaensis* Polaszek sp. n.** urn:lsid:zoobank.org:act:148B35EB-4B8B-44E4-AC56-1F4BC5F7050CORRECT

**Figs 1–11.** Male (holotype): length 0.75 mm.

Colour (holotype): almost entirely dark brown-black, with the following paler (Figs 1 and 2): apices of all tibiae; basitarsi. Wings hyaline.

Morphology (holotype): Vertex smoothly rounded onto occiput; entire vertex with deep reticulate sculpture (Fig 3); occiput entirely smooth, occipital carina present, higher and weaker centrally (Fig 4); hyperoccipital carina absent; frons smooth, reticulate sculpture present between lower inner eye margins and toruli; interantennal process and frontal depression

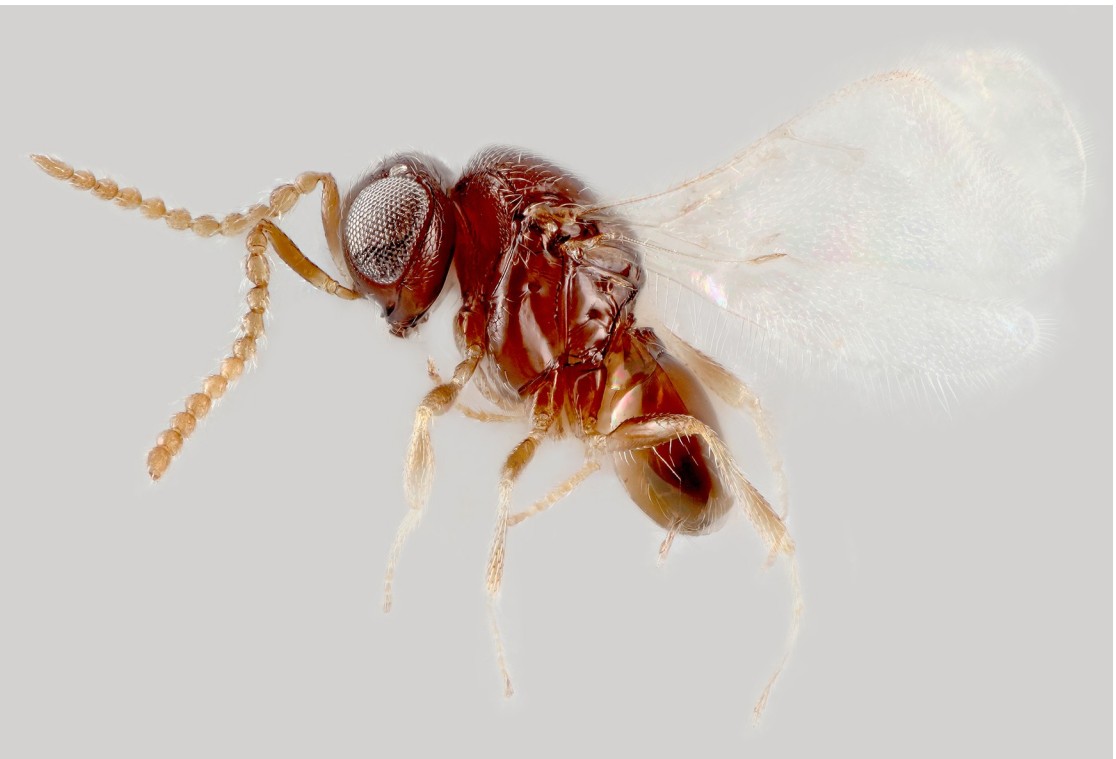

**Fig 1. *Telenomus nizwaensis* Polaszek male holotype.** Lateral habitus.

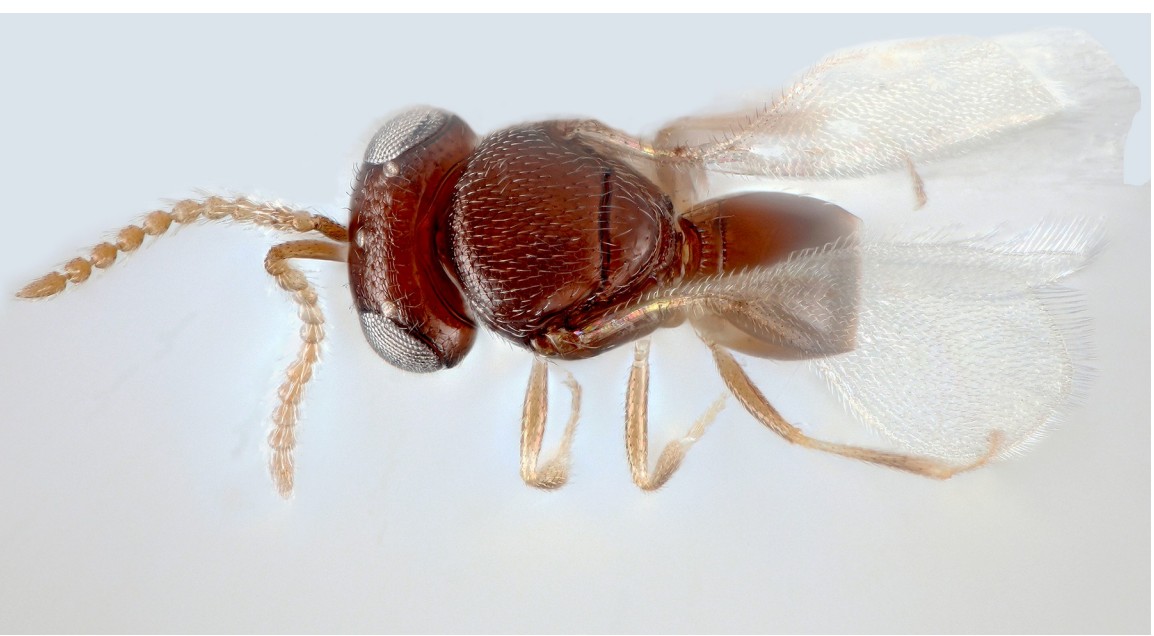

**Fig 2. *Telenomus nizwaensis* Polaszek male holotype.** Dorsal habitus.

absent. Frons moderately convex between inner orbits and toruli; eyes setose; malar region smooth; malar sulcus present; gena with reticulate sculpture behind eyes. Antennae 11-merous.

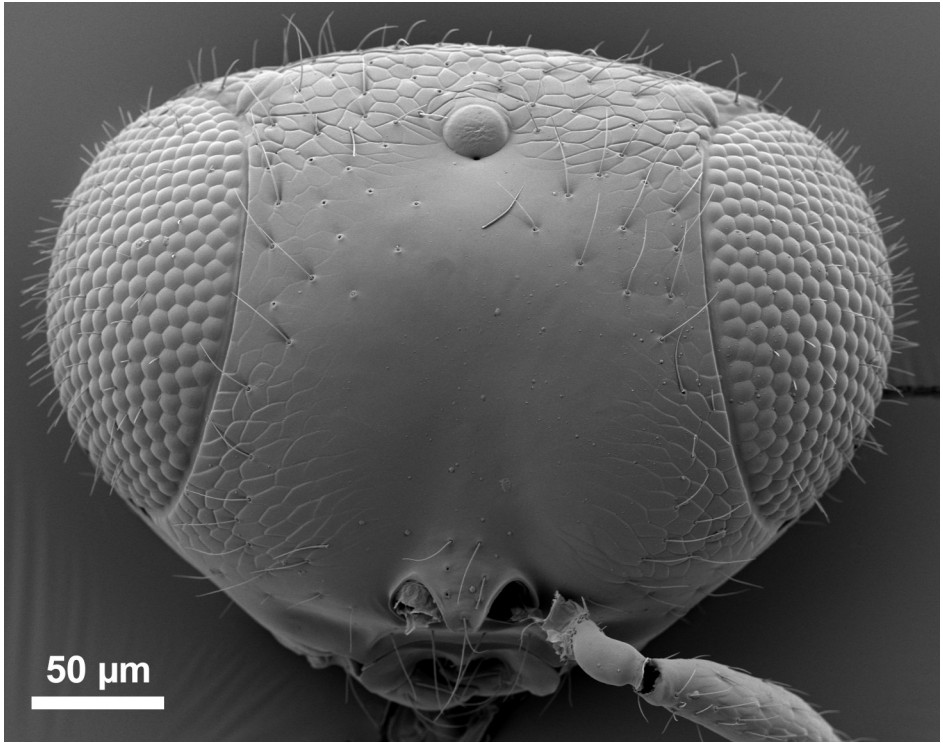

**Fig 3. *Telenomus nizwaensis* Polaszek male paratype.** Face.

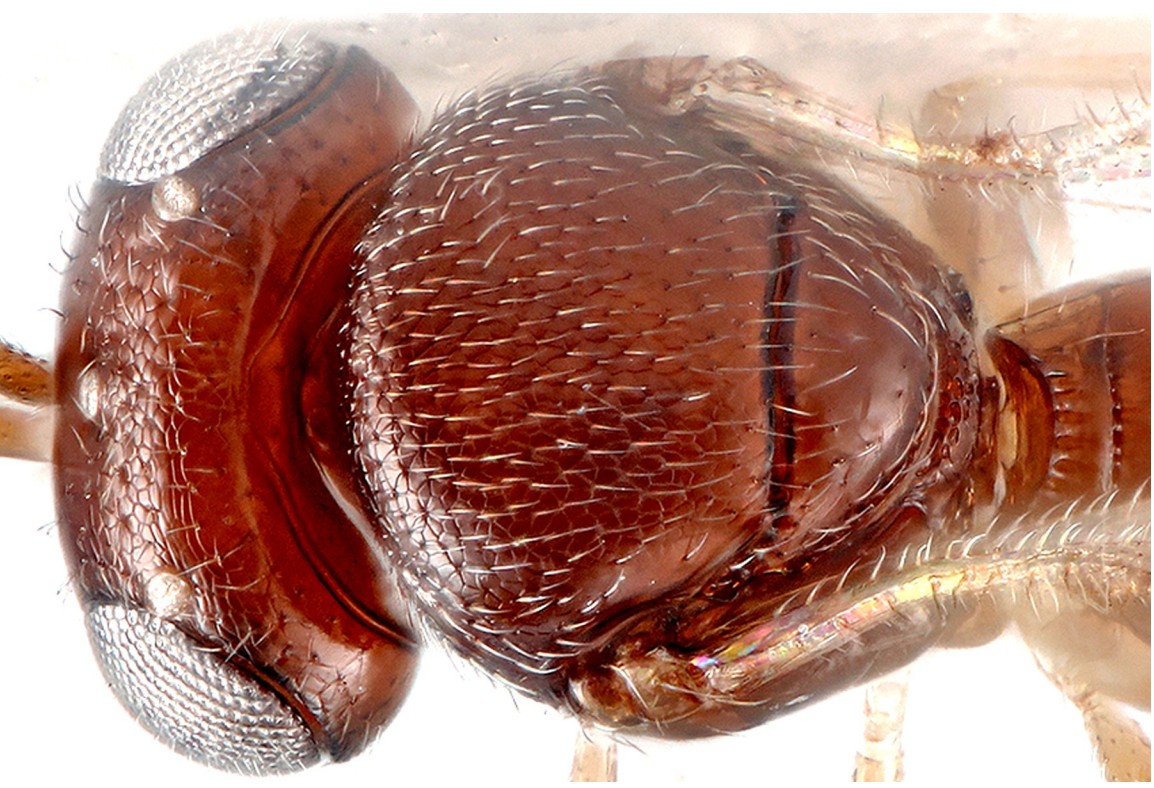

**Fig 4.**

Mesoscutum flattened, entirely reticulately sculptured except posterior lateral corners (above axillae) which are smooth; scutellum mostly smooth; axillae smooth. Lateral portion of scutoscutellar sulcus foveolate (Figs 5 and 6). Length of intercoxal space longer than fore coxae. Netrion (Fig 6) present as area of smooth surface sculpture, delimited dorsally by weak indication of netrion sulcus. Mesopleural pit (mspp Fig 6) deep, slightly transverse, with weak sulcus extending towards tegula. Acropleural sulcus (asu Fig 6) more or less continuous, the foveae almost completely merged; acetabular carina short, without post acetabular sulcus (i.e. no foveae present; ac Fig 4); postacetabular patch (papc Fig 6) clearly indicated, setose; episternal foveae absent; metapleural carina broadly foveolate (mc, Fig 6); metapleural sulcus (mtps Fig 6) indicated as a weak, shallow groove posterior to the metapleural pit (mtpp Fig 6); anteroventral surface of metapleuron with a series of fine grooves that extend towards the acutely pointed anteroventral extension (aem, Fig 6). Metascutellum (= dorsellum) with deep, regular reticulation in anterior half and weak longitudinal carinae in posterior half, interstices smooth (Fig 5).

Metasoma (Fig 1) T1 with 1 pair of sublateral setae, 2 pairs of lateral setae. Basal costae on T1 reaching its posterior margin centrally; basal costae on T2 present as a row of foveae. Ventral metasoma (Fig 7) entirely smooth, a pair of large setae close to the posterior margin of S2; many smaller setae present on laterotergites centrally.

Genitalia (Fig 8). Central projection absent; digiti large relative to aedeagal lobe (about half its length); aedeagal lobe 0.40× length aedeagovolsellar shaft, truncate. Digiti with 3 digital teeth. Basal ring comprising 0.30× length of entire aedeagus.

Variation. Length 0.71–0.79 mm. Extensive variation in colour with many specimens, including paratypes with the metasoma lighter than in the holotype. Morphologically extremely uniform.

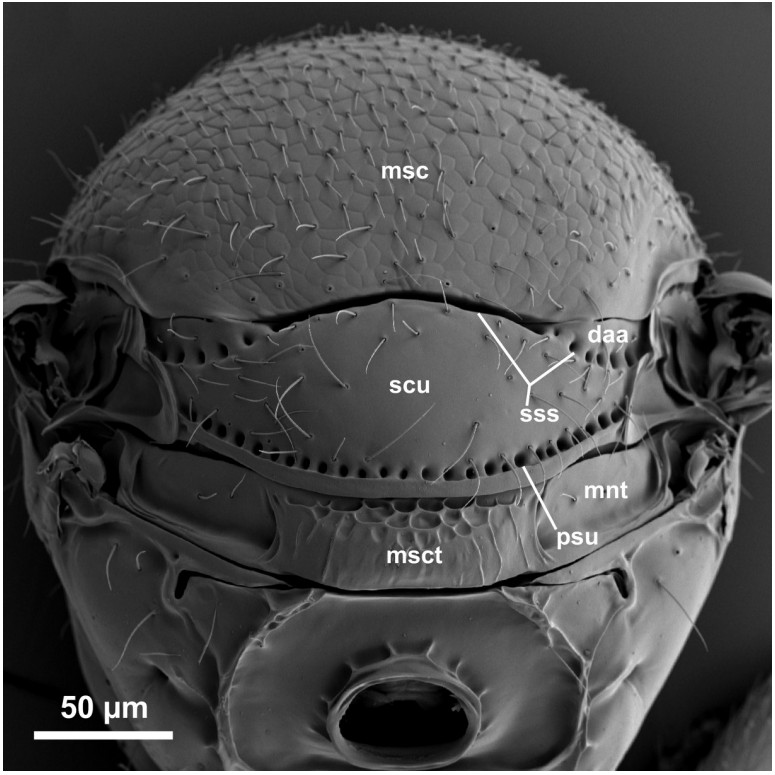

**Fig 5.** *Telenomus nizwaensis* **Polaszek male paratype.** Lateral mesosoma.

Female: Morphologically similar to male with the main exception of the antenna. Clava present and 4-merous, papillary sensilla on A8(1); A9(2); A10(2) A11(1) (Figs 9–11). F1 slightly longer than wide, all other antennomeres wider than long, except A11 (terminal claval segment).

*Species-group placement. Telenomus californicus-group*

*Hosts*. Known from the eggs of its natural host, *D. livia* (Lepidoptera: Lycaenidae). Observations of emergence from eggs of other hosts require confirmation.

*Distribution*. Oman.

Material examined: Holotype ♂: OMAN, Al Jabal Al Akhdar (part of the Hajar mountain range in Ad Dakhiliyah Governorate of Oman; capital city of Nizwa); June 2015 ex eggs of *D. livia*. R. Al Shidi and A. Al-Riyami col. DNA1309: A14; genitalia mounted separately on microscope slide (NHMUK 013377700; HYM 9.1024). Paratypes: 5♀ 20♂, same data as holotype (3♀ [2 sputter-coated for SEM] 10♂, NHMUK 013378214–03378227; 1♀ 1♂ ONHM, 1♀ 1♂ USNM).

## Molecular analyses

Non-destructive extraction and sequencing of three *T. nizawaensis* specimens produced DNA fragments of 414 (28S) and 658 (CO1) nucleotides. The nucleotide sequence of each gene was identical in each of the three specimens. The sequence alignments produced with MAFFT were 475 (28S), 851 (CO1), and 1326 (28S+COI) characters in length (base pairs plus gaps). In all phylogenetic analyses, the three specimens of *T. nizwaensis* clustered together unequivocally with 100% bootstrap support (Fig 12). Sequence data associated with each specimen has been deposited in GenBank under accession numbers MT635051–MT635053 (CO1) and MT636558–MT636560 (28S).

**28S D2 (nuclear ribosomal).** A blastn search of the *T. nizwaensis* 28S nucleotide sequence against the NCBI nt database produced two hits identical of max score (712) and percent identity (97.8) to undetermined *Telenomus* spp. (GenBank accessions JX683253 and JX683244). The closest described species with which *T. nizwaensis* matched was *T. sechellensis* (703 max score; 97.3% identity). *Telenomus nizwaensis* was recovered as sister to a grouping of *T. goniopis*, an unidentified *Telenomus*, *T. busseolae*, and *T. sechellensis*.

**CO1 "barcode" (mitochondrial).** A blastn search of the *T. nizwaensis* CO1 nucleotide sequence against the NCBI nt database yielded best matches to *Telenomus dignus* KR270640 (859 max score), and Scelionidae sp. KM566105 and Scelioninae sp. HM883306 (92.1% identity). The phylogenetic analysis based on the CO1 sequences of *T. nizwaensis* and eight other species of *Telenomus* resulted in a sister group relationship between *T. nizwaensis* and *T. consimilis* + *T.dolichocerus*, albeit with low bootstrap support.

## Discussion

### Taxonomy

There are 628 described, extant species of *Telenomus*. Synonymies are therefore extremely likely to be detected in future, as happened recently (and spectacularly) in the closely related

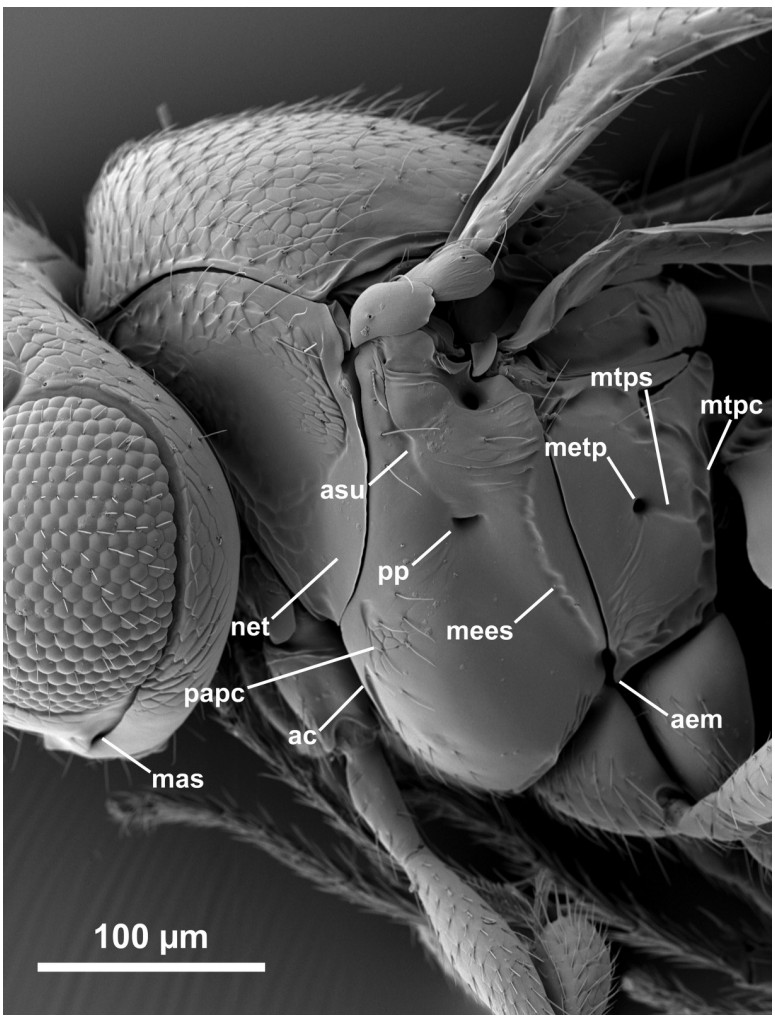

**Fig 6.** *Telenomus nizwaensis* **Polaszek male paratype.** Dorsal mesosoma.

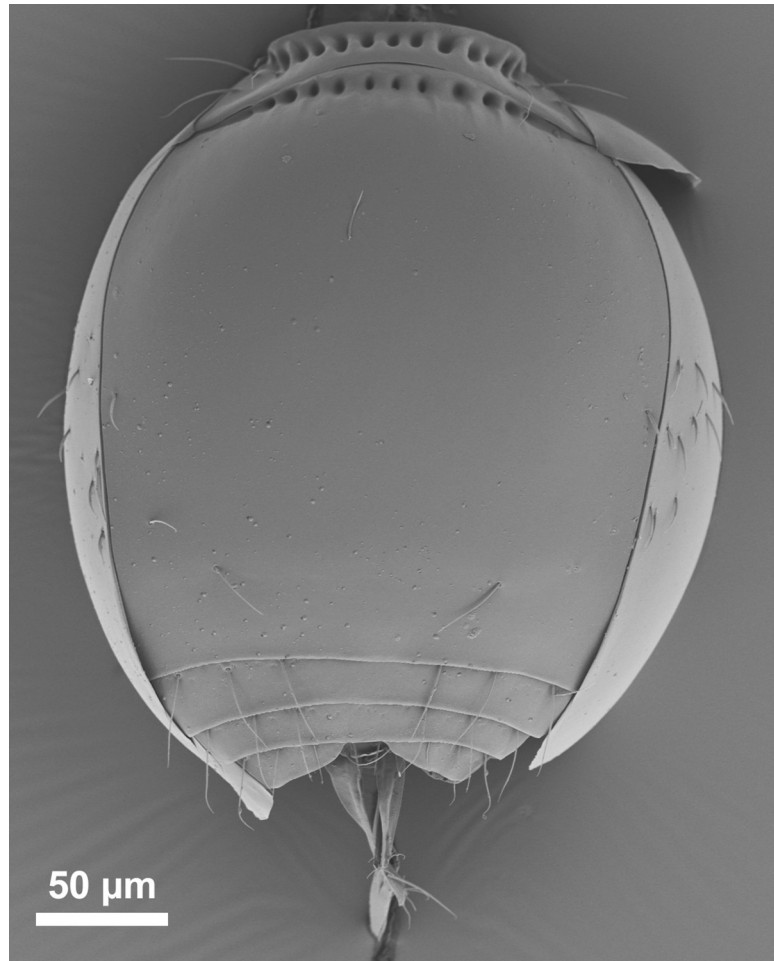

**Fig 7. *Telenomus nizwaensis* Polaszek female paratype.** Ventral metasoma.

genus *Trissolcus* [10]. The possibility that *T. nizwaensis* was described previously under another name cannot be ruled out. However, the task of obtaining, or attempting to obtain, each *Telenomus* species holotype to exclude that possible species identity, would at least severely delay, and possibly prohibit, the execution of this work. The morphological and molecular data presented here permit the unequivocal identification of *T. nizwaensis*. Future studies, especially using newly developed technologies, may discover a senior synonym of *T. nizwaensis*.

The 4-merous clava in the female has not been observed previously in any *Telenomus* species of the *T. californicus* goup, all members of which have been assumed to have the clava 5-merous [9]. However, papillary sensilla can be extremely difficult to see, and the presence of a single sensillum on A7 may have been wrongly assumed in the past for some species.

Guided in part by the extensive collections of Old World *Telenomus* at NHMUK, together with groundwork provided by Johnson [9], it has been possible to eliminate all African and Asian species described by Nixon [24–30], as well as many others held in the NHMUK collection. There remain a great many species of *Telenomus* described by Kozlov and Konoova [31], some from Central Asia that could conceivably also be present in the Arabian Peninsula.

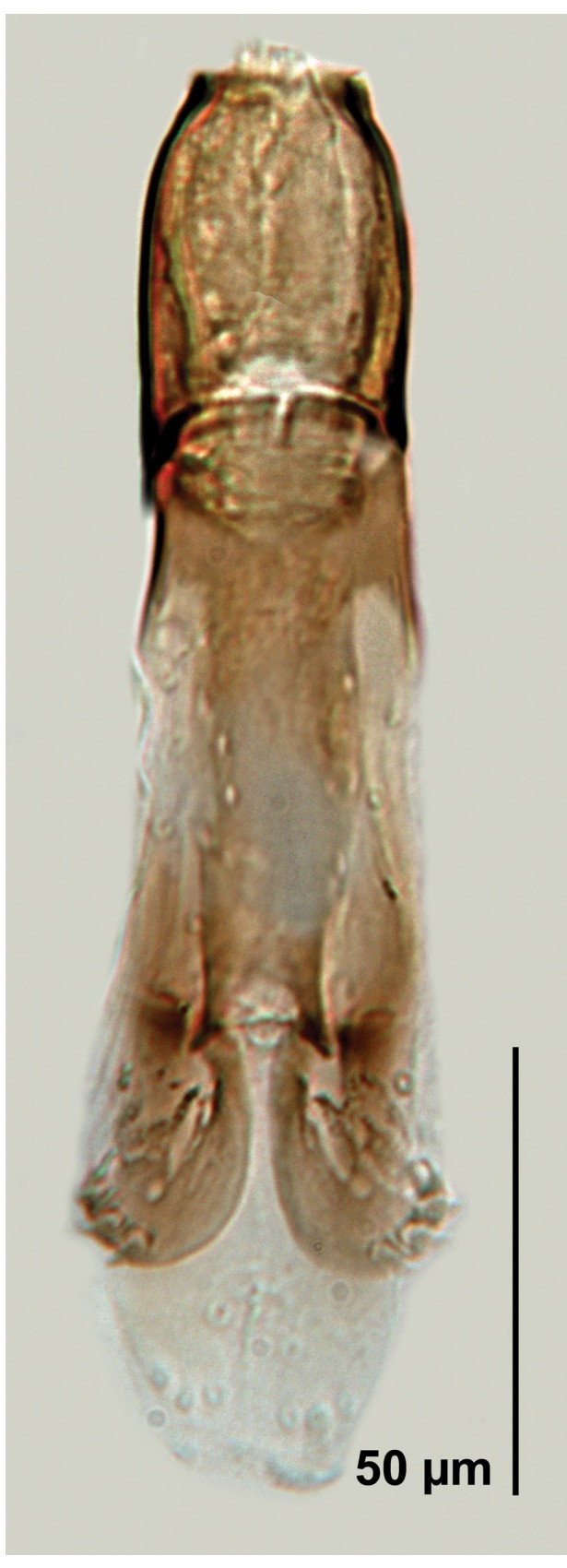

**Fig 8.** *Telenomus nizwaensis* **Polaszek male genitalia.**

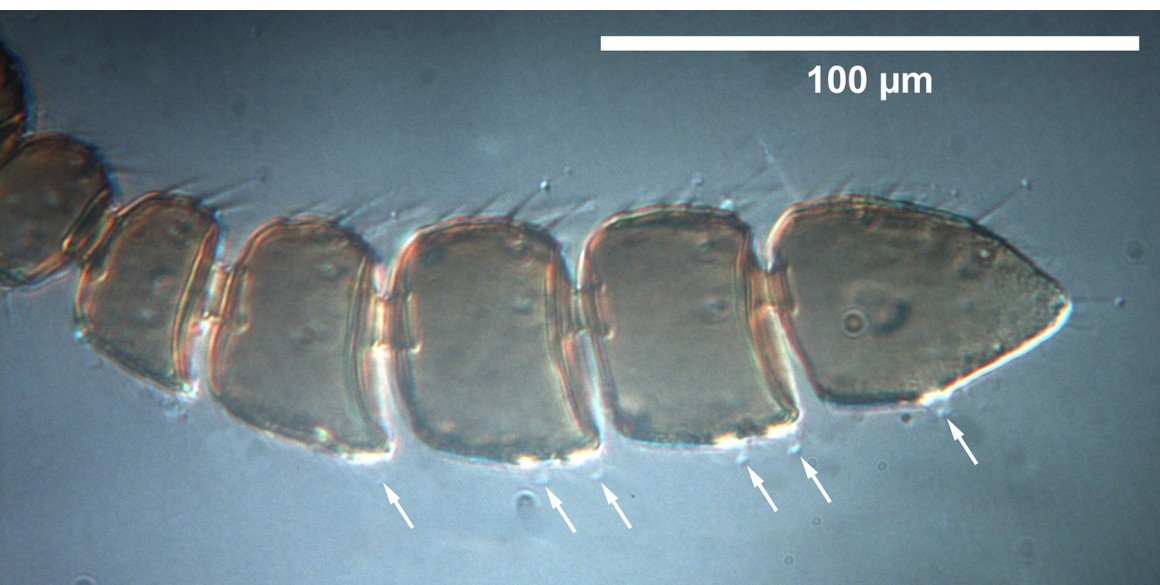

**Fig 9. *Telenomus nizwaensis* Polaszek female paratype.** Terminal antennomeres under differential interference contrast. Papillary sensilla indicated by arrows.

### Biology and economic importance

*Telenomus* is a highly speciose genus of minute wasps in the family Scelionidae (included until recently in the family Platygastridae), all of which are endoparasitoids of eggs of several insect orders, including the Heteroptera, Diptera, Neuroptera and, especially, the Lepidoptera. Some species, such as *Telenomus remus*, the most common egg parasitoid of *Spodoptera frugiperda* in the Americas, are very well-known parasitoid and are mass-reared and deployed in augmentative biological control in many countries [32]. *T. nizwaensis* is not well known but it is a natural enemy of a pest of economic importance, and has clear potential as an indigenous agent of biological control in Oman, and may also prove effective elsewhere.

Field notes and samples collected from *D. livia* eggs suggest that its life-history is similar to other species in the subfamily Telenominae. For instance, females of *Telenomus fariai* start

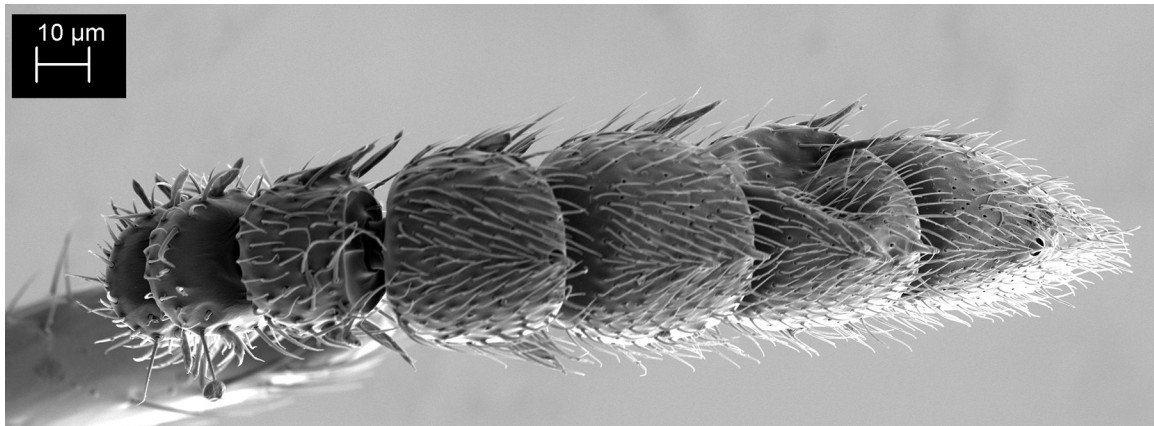

**Fig 10. *Telenomus nizwaensis* Polaszek female.** Scanning electron micrograph of terminal antennomeres showing papillary sensilla present on A8-11 and clearly absent from A7.

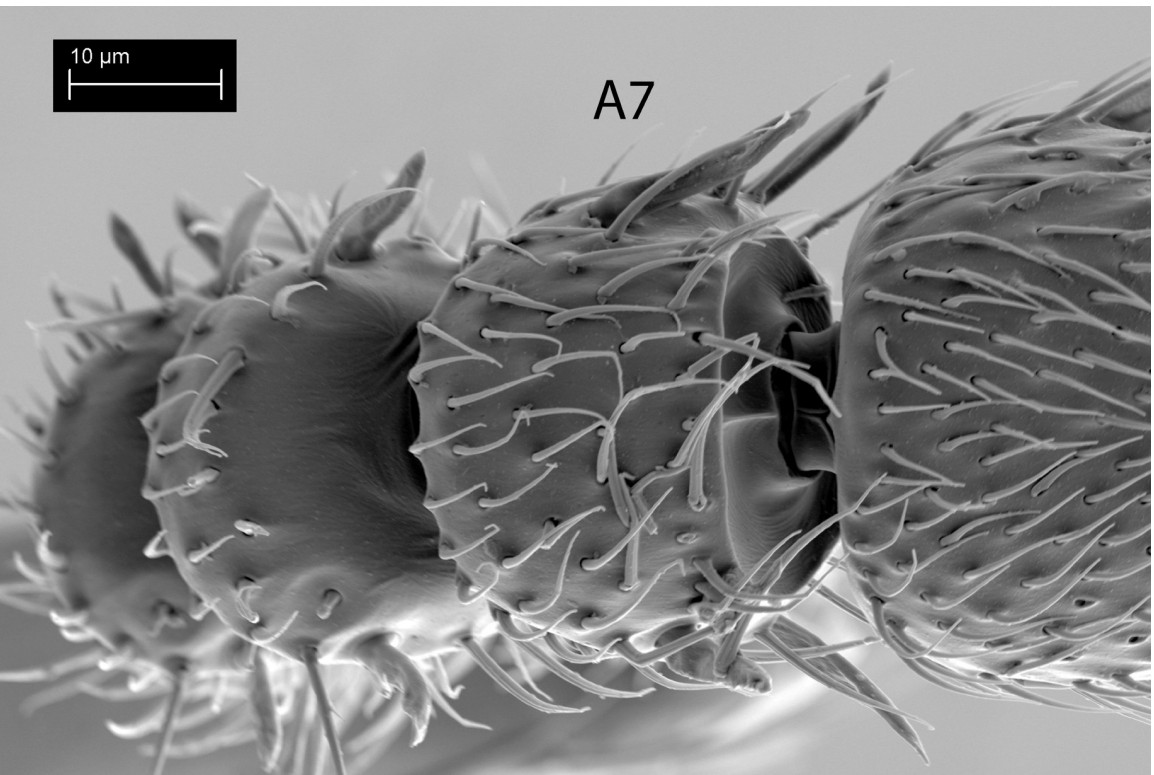

**Fig 11. *Telenomus nizwaensis* Polaszek female.** Detail of A7 showing lack of papillary sensilla.

ovipositing hours after emergence [33] and similarly, newly emerged *T. nizwaensis* females attack fresh eggs of the host, *D. livia* (A. A-R. pers. obs.). *T. nizwaensis* is a solitary species: females lay a single egg per host egg, or at least only one adult emerges from each host, as reported for *T. remus* [34]. *Telenomus* adult usually emerge through a hole in the side of the host's chorion [35] and similar emergence holes are observed for *T. nizwaensis*.

A constraint in studying *T. nizwaensis* is that successful laboratory rearing protocols have not yet been developed. Eggs of wild tiger moth *Utetheisa* sp., cabbage looper *Trichoplusia ni*, and cotton leafworm *Spodoptera littoralis* have been exposed to *T. nizwaensis* parasitoids by researchers in the Omani Directorate General of Agricultural and Livestock Researches of the Ministry of Agriculture and Fisheries (2014), but no *T. nizwaensis* emerged from any of these eggs. *Deudorix livia* migrates to *Acacia* sp. (locally called Talh) during mid-September, at the end of the pomegranate season, and *Telenomus* wasps that may have been *T. nizwaensis* have emerged from butterfly eggs infesting *Acacia* sp. pods (A. A-R. pers. obs.).

Current field data indicate that *T. nizwaensis* is the most dominant parasitoid of *D. livia* in the Al Jabal Al Akhdar region. Annual reports of the IPM program run by the Omani Directorate General of Agricultural and Livestock Researches of the Ministry of Agriculture and Fisheries show that the parasitism rate by *T. nizwaensis* over the past 15 years was 62%, compared with 8.2% by *Trichogramma* sp. *Telenomus nizwaensis* begins attacking *D. livia* later in the pomegranate fruiting season, with peak parasitism between mid-June and mid-July, when pomegranate trees start to flower at the end of April. Alternative control measures are likely to be needed if earlier damage by *D. livia* is to be avoided. Sex ratios of *T. nizwaensis* emerging from naturally collected *D. livia* eggs in Oman (temperature range 20–35°C, June-July 2020)

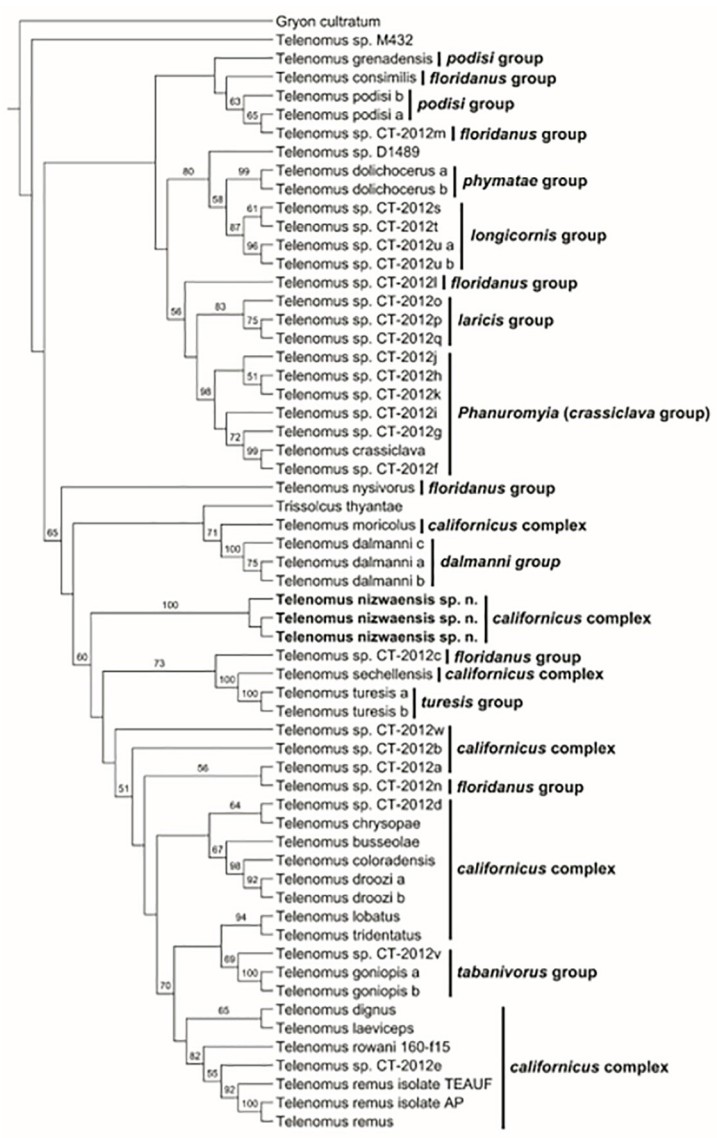

**Fig 12. RAxML analyses (CO1, 28S, CO1+28S).**

have a female biased sex ratio (proportion of offspring that are male ≈ 0.37, A. Al-R. unpublished data). Female biased sex ratios are generally a positive attribute in term of a parasitoid's ability to suppress pest populations because only the female adults attack the next generation of hosts, and fewer host resources were used for the production of males [36].

## Conclusions

We have formally described a new species of scelionid wasp, *Telenomus nizwaensis* Polaszek. Its biology is similar to that of its congeners and, as a natural enemy of the pomegranate butterfly, *D. livia*, it is a beneficial component of pomegranate agro-ecosystems in Oman. Current studies are aimed at further evaluating and promoting its field performance as an agent of biological pest control.

## Supporting information

**S1 Fig.**
(TIF)

**S2 Fig.**
(TIF)

**S1 Table. Taxa, GenBank accession numbers, species group placement, collection location, and references associated with the sequence data used in the phylogenetic analyses.** Species group/complex placement mainly follows Johnson (1984) and Taekul et al. (2013).
(XLSX)

**S1 Movie.**
(MP4)

## Acknowledgments

We thank the Ministry of Agriculture, Fisheries and Water Resources Agricultural Wealth, Fisheries and Water Resources in Oman for support, especially Naser Al Abri and Issa Al-Mandhari for their help in field work while collection and for offering other essential logistics. We also thank the Department of Agricultural Development in Al Jabal Al Akhdar for their valuable help, in particular as the contact point between us and local farmers.

## Author Contributions

**Conceptualization:** A. Polaszek, A. Al-Riyami, S. A. Al-Khatri, R. H. Al-Shidi, I. C. W. Hardy.

**Data curation:** A. Polaszek.

**Formal analysis:** A. Polaszek, Z. Lahey.

**Investigation:** A. Polaszek, R. H. Al-Shidi.

**Methodology:** A. Polaszek.

**Resources:** A. Polaszek, I. C. W. Hardy.

**Writing – original draft:** A. Polaszek, Z. Lahey, I. C. W. Hardy.

**Writing – review & editing:** A. Polaszek, A. Al-Riyami, Z. Lahey, I. C. W. Hardy.

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
