## [Decision Letter · Decision Letter 0]

7 Jan 2021

PONE-D-20-34715

Telenomus nizwaensis (Hymenoptera: Scelionidae), an important egg parasitoid of the pomegranate butterfly Deudorix livia Klug (Lepidoptera: Lycaenidae) in Oman

PLOS ONE

Dear Dr. Polaszek,

Thank you for submitting your manuscript to PLOS ONE. After careful consideration, we feel that it has merit but does not fully meet PLOS ONE’s publication criteria as it currently stands. Therefore, we invite you to submit a revised version of the manuscript that addresses the points raised during the review process.

We look forward to receiving your revised manuscript.

Kind regards,

Feng ZHANG, Ph.D.

Academic Editor

PLOS ONE

Journal Requirements:

2. We noticed you have some minor occurrence of overlapping text with the following previous publication, which needs to be addressed:

https://journals.plos.org/plosone/article?id=10.1371%2Fjournal.pone.0223761

In your revision ensure you cite all your sources (including your own work), and quote or rephrase any duplicated text outside the methods section. Further consideration is dependent on these concerns being addressed.

3. In your Methods section, please provide additional location information of the collection sites, including geographic coordinates for the data set if available.

"We thank the Ministry of Agriculture, Fisheries and Water Resources Agricultural Wealth,

304 Fisheries and Water Resources in Oman for support, especially Naser Al Abri and Issa Al-Mandhari

305 for their help in field work while collection and for offering other essential logistics. A. A-R was

306 funded by the Sultanate of Oman, Ministry of Higher Education, Research and Innovation under

307 grant number PGE055580 D.P.S/321/2017. We thank the Anglo-Omani Society for assistance

8

with publication charges. We also thank the Department of 308 Agricultural Development in Al Jabal

309 Al Akhdar for their valuable help, in particular as the contact point between us and local farmers."

"A. A-R: PGE055580 D.P.S/321/2017

Sultanate of Oman, Ministry of Higher Education, Research and Innovation

https://www.mohe.gov.om

Reviewers' comments:

Reviewer's Responses to Questions

**Comments to the Author**

1. Is the manuscript technically sound, and do the data support the conclusions?

Reviewer #1: Yes

Reviewer #2: Yes

2. Has the statistical analysis been performed appropriately and rigorously? 

Reviewer #1: N/A

Reviewer #2: N/A

3. Have the authors made all data underlying the findings in their manuscript fully available?

Reviewer #1: Yes

Reviewer #2: Yes

4. Is the manuscript presented in an intelligible fashion and written in standard English?

Reviewer #1: Yes

Reviewer #2: Yes

5. Review Comments to the Author

Reviewer #1: In this manuscript, the authors describe a new species, Telenomus nizwaensis Polaszek sp. n., based on morphology and DNA sequence data. This egg parasitoid is an important biological agent against the pomegranate butterfly Deudorix (=Virachola) livia, which is a major pest of pomegranate in Oman. They also summarize the biology of the parasitoid species, which might enhance to the further use of this parasitoid in biological control programs. This study is quite straightforward and the manuscript is well written. There are a few points of consideration listed below that I think might improve the manuscript.

Major comments

1. In the method section, the authors say that eight individuals (4 females, 4 males) were subjected to DNA extraction (line 106), while in the result section they mentioned that 28S and COI were generated only from three specimens (line 208). So only these three specimens were successfully sequenced? What are the genders of these three specimens? It would be good to know that if these three specimens cover both sexes. Apparently, the studied specimens were reared from the eggs of the pomegranate butterfly from the same locality and it is mostly likely, the emerged parasitoids belong to the same species, but you never know, other species of Telenomus might present, too. If the sequenced specimens are both from female and male, then the association of both sexes can be sure.

2. The new species is placed in the Telenolmus californicus-group, I believe it would be useful to readers if the authors could provide more information on the diagnosis of this species group. Besides, since the new species has a 4-merous clava in the female, which is different from other species of the group, the placement in the group should be discussed more in more details. In other words, why do the authors believe this new species belong to the Telenolmus californicus-group?

3. The author stated that the CO1 sequence of the new species is best match to Telenomus dingus KR270640 (line 225), what is the identical percentage between these two species? And why the CO1 sequence of Telenomus dingus KR270640 was not included in the phylogenetic analysis?

4. Line 95-96, the authors state “Observations of emergence from eggs of other hosts require confirmation”, also line 277-288 “Telenomus wasps that may have been T. nizwaensis have emerged from butterfly eggs infesting Acacia sp. pods (A. A-R. pers. obs.)”. The confirmation of the host range of the new species is important to biological control programs, it would be better to include the DNA of specimens emerged from other hosts into the analysis. Of course, this is optional if there were no such specimens available.

Minor comments:

1. In figs 12-14, the first two trees indicated as RAxML, while the last is not, is it also RAxML?

2. Please provide the full descriptions for the abbreviations in figs 5-6, either in the main text or the legend.

3. It seems that there are duplicates of figs 9-10.

Reviewer #2: Telenomus is a large and difficult genus to study as the authors mentioned. The authors described a new parasitoid species with host information and DNA barcodes. The description of this new hymenopteran species will benefit the management of its host pest as a potential biological control agent, as well as facilitate the species delimitations/identifications of the genus Telenomus. This new species is placed in the californicus-group, however, the interpretation of this species group was not provided. Further, the sister grouping relationships were recovered in the phylogenetic analyses with limited data. However, the possibly close relationships of this new species with another species in the californicus-group were not discussed. The discussions about its close relationship with other species and the interpretation of the species group where it is placed could help readers to understand the systematic position of this species in this species group or even in this genus.

Please see the details as in the attached file.

6. PLOS authors have the option to publish the peer review history of their article (what does this mean?). If published, this will include your full peer review and any attached files.

Reviewer #1: No

Reviewer #2: No

---

## [Author Response · Author response to Decision Letter 0]

23 Feb 2021

PONE-D-20-34715

Telenomus nizwaensis (Hymenoptera: Scelionidae), an important egg parasitoid of the pomegranate butterfly Deudorix livia Klug (Lepidoptera: Lycaenidae) in Oman

PLOS ONE

RESPONSE TO REVIEWERS (our responses in blue text)

2. We noticed you have some minor occurrence of overlapping text with the following previous publication, which needs to be addressed:

https://journals.plos.org/plosone/article?id=10.1371%2Fjournal.pone.0223761

In your revision ensure you cite all your sources (including your own work), and quote or rephrase any duplicated text outside the methods section. Further consideration is dependent on these concerns being addressed.

• Since the extraction + sequencing protocols, several imaging protocols and other aspects of the methods were exactly identical to those in previous publications, we see no reason to artificially modify them for the sake of making them read differently. However, we acknowledge that the previous +/- identical protocols should be cited. At the risk of over self-citation I have cited only https://doi.org/10.1371/journal.pone.0223761 (Goniozus omanensis) and not https://doi.org/10.1371/journal.pone.0230944 (Metaphycus macadamiae). Unfortunately this has necessitated re-numbering the references.

• 

3. In your Methods section, please provide additional location information of the collection sites, including geographic coordinates for the data set if available.

• Done

"We thank the Ministry of Agriculture, Fisheries and Water Resources Agricultural Wealth,

304 Fisheries and Water Resources in Oman for support, especially Naser Al Abri and Issa Al-Mandhari

305 for their help in field work while collection and for offering other essential logistics. A. A-R was

306 funded by the Sultanate of Oman, Ministry of Higher Education, Research and Innovation under

307 grant number PGE055580 D.P.S/321/2017. We thank the Anglo-Omani Society for assistance

8

with publication charges. We also thank the Department of 308 Agricultural Development in Al Jabal

309 Al Akhdar for their valuable help, in particular as the contact point between us and local farmers."

"A. A-R: PGE055580 D.P.S/321/2017

Sultanate of Oman, Ministry of Higher Education, Research and Innovation

https://www.mohe.gov.om

• I have removed the specific funding references for the Oman Ministry. Should I also do this for the publication costs?

• We have submitted the unaligned and aligned fasta files to Dryad. Those files can be accessed at the following link: https://doi.org/10.5061/dryad.sxksn0329.

• This should fine – I have been lead/corres[poning author on 2 PLoSOne papers since 2016

• Done

Reviewers' comments:

Reviewer's Responses to Questions

Comments to the Author

1. Is the manuscript technically sound, and do the data support the conclusions?

Reviewer #1: Yes

Reviewer #2: Yes

2. Has the statistical analysis been performed appropriately and rigorously? 

Reviewer #1: N/A

Reviewer #2: N/A

3. Have the authors made all data underlying the findings in their manuscript fully available?

Reviewer #1: Yes

Reviewer #2: Yes

4. Is the manuscript presented in an intelligible fashion and written in standard English?

Reviewer #1: Yes

Reviewer #2: Yes

5. Review Comments to the Author

Reviewer #1: In this manuscript, the authors describe a new species, Telenomus nizwaensis Polaszek sp. n., based on morphology and DNA sequence data. This egg parasitoid is an important biological agent against the pomegranate butterfly Deudorix (=Virachola) livia, which is a major pest of pomegranate in Oman. They also summarize the biology of the parasitoid species, which might enhance to the further use of this parasitoid in biological control programs. This study is quite straightforward and the manuscript is well written. There are a few points of consideration listed below that I think might improve the manuscript.

Major comments

1. In the method section, the authors say that eight individuals (4 females, 4 males) were subjected to DNA extraction (line 106), while in the result section they mentioned that 28S and COI were generated only from three specimens (line 208). So only these three specimens were successfully sequenced? What are the genders of these three specimens? It would be good to know that if these three specimens cover both sexes. Apparently, the studied specimens were reared from the eggs of the pomegranate butterfly from the same locality and it is mostly likely, the emerged parasitoids belong to the same species, but you never know, other species of Telenomus might present, too. If the sequenced specimens are both from female and male, then the association of both sexes can be sure.

This information has now been updated

2. The new species is placed in the Telenolmus californicus-group, I believe it would be useful to readers if the authors could provide more information on the diagnosis of this species group. Besides, since the new species has a 4-merous clava in the female, which is different from other species of the group, the placement in the group should be discussed more in more details. In other words, why do the authors believe this new species belong to the Telenolmus californicus-group?

• This point is very well-made. One of us (ZL) has thoroughly updated the discussion on the species-group placement after discussion with Prof. N. Johnson., the author of the “californicus complex”

3. The author stated that the CO1 sequence of the new species is best match to Telenomus dingus KR270640 (line 225), what is the identical percentage between these two species? And why the CO1 sequence of Telenomus dingus KR270640 was not included in the phylogenetic analysis?

• The analyses have been completed re-done. We included all available and appropriate sequnnces from GenBank as well as the sited sequence from Telenomus dignus, though that had to be extracted from a mitogenomic study. The results are much more detailed and rigorous, although conclusions from the first analysis are still completely supported. We have added a list of all sequences analysed, their depositories etc as supplementary files.

4. Line 95-96, the authors state “Observations of emergence from eggs of other hosts require confirmation”, also line 277-288 “Telenomus wasps that may have been T. nizwaensis have emerged from butterfly eggs infesting Acacia sp. pods (A. A-R. pers. obs.)”. The confirmation of the host range of the new species is important to biological control programs, it would be better to include the DNA of specimens emerged from other hosts into the analysis. Of course, this is optional if there were no such specimens available.

• Yes, that would me a “nice to have”, but considering how many 100s of undescribed Telenomus species there must be out there, simply not practically possible right now. At the very least, our thorough analysis shows that T. nizwaenesis does not cluster closely with any other known species.

Minor comments:

1. In figs 12-14, the first two trees indicated as RAxML, while the last is not, is it also RAxML?

2. Please provide the full descriptions for the abbreviations in figs 5-6, either in the main text or the legend.

3. It seems that there are duplicates of figs 9-10.

• All these points have been addressed

Reviewer #2: Telenomus is a large and difficult genus to study as the authors mentioned. The authors described a new parasitoid species with host information and DNA barcodes. The description of this new hymenopteran species will benefit the management of its host pest as a potential biological control agent, as well as facilitate the species delimitations/identifications of the genus Telenomus. This new species is placed in the californicus-group, however, the interpretation of this species group was not provided. Further, the sister grouping relationships were recovered in the phylogenetic analyses with limited data. However, the possibly close relationships of this new species with another species in the californicus-group were not discussed. The discussions about its close relationship with other species and the interpretation of the species group where it is placed could help readers to understand the systematic position of this species in this species group or even in this genus.

• All points well-made and addressed thoroughly in our revision – see also the points able

Please see the details as in the attached file.

6. PLOS authors have the option to publish the peer review history of their article (what does this mean?). If published, this will include your full peer review and any attached files.

Do you want your identity to be public for this peer review? For information about this choice, including consent withdrawal, please see our Privacy Policy.

Reviewer #1: No

Reviewer #2: No

---

## [Decision Letter · Decision Letter 1]

7 Apr 2021

Telenomus nizwaensis (Hymenoptera: Scelionidae), an important egg parasitoid of the pomegranate butterfly Deudorix livia Klug (Lepidoptera: Lycaenidae) in Oman

PONE-D-20-34715R1

Dear Dr. Polaszek,

We’re pleased to inform you that your manuscript has been judged scientifically suitable for publication and will be formally accepted for publication once it meets all outstanding technical requirements.

Kind regards,

Feng ZHANG, Ph.D.

Academic Editor

PLOS ONE

Additional Editor Comments (optional):

Reviewers' comments:

Reviewer's Responses to Questions

**Comments to the Author**

1. If the authors have adequately addressed your comments raised in a previous round of review and you feel that this manuscript is now acceptable for publication, you may indicate that here to bypass the “Comments to the Author” section, enter your conflict of interest statement in the “Confidential to Editor” section, and submit your "Accept" recommendation.

Reviewer #1: All comments have been addressed

Reviewer #2: (No Response)

2. Is the manuscript technically sound, and do the data support the conclusions?

Reviewer #1: Yes

Reviewer #2: Yes

3. Has the statistical analysis been performed appropriately and rigorously? 

Reviewer #1: N/A

Reviewer #2: Yes

4. Have the authors made all data underlying the findings in their manuscript fully available?

Reviewer #1: Yes

Reviewer #2: Yes

5. Is the manuscript presented in an intelligible fashion and written in standard English?

Reviewer #1: Yes

Reviewer #2: Yes

6. Review Comments to the Author

Reviewer #1: (No Response)

Reviewer #2: (No Response)

7. PLOS authors have the option to publish the peer review history of their article (what does this mean?). If published, this will include your full peer review and any attached files.

Reviewer #1: No

Reviewer #2: No

---

## [Editor Report · Acceptance letter]

12 Apr 2021

PONE-D-20-34715R1 

*Telenomus nizwaensis* (Hymenoptera: Scelionidae), an important egg parasitoid of the pomegranate butterfly *Deudorix livia* Klug (Lepidoptera: Lycaenidae) in Oman 

Dear Dr. Polaszek:

I'm pleased to inform you that your manuscript has been deemed suitable for publication in PLOS ONE. Congratulations! Your manuscript is now with our production department. 

Kind regards, 

on behalf of

Dr. Feng ZHANG 

Academic Editor

PLOS ONE